# Characterizing the Impact of Communication on Cellular and Collective Behavior Using a Three-Dimensional Multiscale Cellular Model

**DOI:** 10.3390/e25020319

**Published:** 2023-02-09

**Authors:** Moriah Echlin, Boris Aguilar, Ilya Shmulevich

**Affiliations:** 1Prostate Cancer Research Center, Faculty of Medicine and Health Technology, Tampere University, FI-33014 Tampere, Finland; 2Institute for Systems Biology, Seattle, WA 98109, USA

**Keywords:** cell–cell communication, Boolean networks, complex systems, multiscale modeling, mathematical biology

## Abstract

Communication between cells enables the coordination that drives structural and functional complexity in biological systems. Both single and multicellular organisms have evolved diverse communication systems for a range of purposes, including synchronization of behavior, division of labor, and spatial organization. Synthetic systems are also increasingly being engineered to utilize cell–cell communication. While research has elucidated the form and function of cell–cell communication in many biological systems, our knowledge is still limited by the confounding effects of other biological phenomena at play and the bias of the evolutionary background. In this work, our goal is to push forward the context-free understanding of what impact cell–cell communication can have on cellular and population behavior to more fully understand the extent to which cell–cell communication systems can be utilized, modified, and engineered. We use an in silico model of 3D multiscale cellular populations, with dynamic intracellular networks interacting via diffusible signals. We focus on two key communication parameters: the effective interaction distance at which cells are able to interact and the receptor activation threshold. We found that cell–cell communication can be divided into six different forms along the parameter axes, three asocial and three social. We also show that cellular behavior, tissue composition, and tissue diversity are all highly sensitive to both the general form and specific parameters of communication even when the cellular network has not been biased towards that behavior.

## 1. Introduction

Communication between cells is a key property of living systems, from unicellular microbes to multicellular organisms. By signaling to each other, cells are able to coordinate their activity, thereby attaining structural and functional complexity. Fundamentally, these complexities are achieved by regulating which cellular behaviors occur within a population and how those behaviors are organized. To achieve this, organisms have evolved communication systems that are diverse in both mechanism and function.

In microbial populations, cell–cell communication systems are pervasive and utilized for a variety of population functions. For example, signaling is used to propagate information through a population to provide long range coordination, as in the use of ion channels to regulate metabolic states during growth cycles in biofilms [1,2]. It can be used to aggregate environmental information to execute a singular collective behavior, such as infectious bacteria like *V. cholera* and *P. aeruginosa* relying on quorum sensing to collectively express virulence factors when pathogenicity will be maximized [3,4]; motile bacteria like *E. coli* secreting a chemical trail to collectively migrate toward food sources [5]; and sporulating bacteria like *B. subtilis* collectively sporulating in response to quorum signals when there is not enough food for colony survival [6]. Cell–cell signaling can also be used to coordinate subpopulations with different and often mutually exclusive functions, such as in the generation of matrix- and surfactant-producing cells in *B. subtilis* colonies during colony expansion [7,8]; and the regular spacing of nitrogen-fixing heterocysts and photosynthetic vegetative cells in filamentous cyanobacteria for survival in low-nitrogen conditions [9,10]. Further examples of microbial communication systems can be seen in colony morphogenesis or multispecies biofilm maintenance [11,12,13].

In multicellular organisms, cell–cell communication is so vital that it is recognized as a defining feature of multicellularity [14]. Indeed, it has been proposed that an expansion in the genes dedicated to cell–cell communication accompanied the transition from unicellular to multicellular life [15]. As in microbial systems, cell–cell signaling in multicellular organisms can serve to synchronize cells in a common behavior or manage diverse subpopulations; however, there is much more breadth and complexity in multicellular communication systems. From simple to complex organisms, across evolutionary kingdoms, cell–cell communication is heavily involved in nearly all multicellular biological operations. During development, multicellular organisms rely on cell–cell communication cues for functions like temporal and spatial cellular differentiation (plant stem morphogenesis [16], nematode neuronal fate decisions [17]); directional cell migration (zebrafish organogenesis [18], mammalian vessel formation [19]); and controlled cell death (avian limb development [20], insect body segmentation [21]). In mature organisms, homeostasis is maintained by cell–cell communication through functions like population size control (murine T cells [22] and macrophage/fibroblasts [23] populations), maintaining multicellular structures (animal corneal maintenance [24] and bone remodeling [25]), and coordination between subsystems (human wound healing [26]). Mature organisms must also navigate their environment as a cohesive unit, using cell–cell communication to perform the complex functions necessary for survival (mollusc neuromuscular coordination [27] and vertebrate adaptive immunity [28]).

In addition to naturally occurring systems, cell–cell communication is increasingly being utilized as a tool in synthetic systems. Many of the communication mechanisms used in living systems can be adapted for synthetic systems. Current research areas include the synthesis of populations capable of swarming behavior [29], reliable spatial patterning [30], organization into multicellular structures [31], biological computing [32], and complex functions like biofuel production [33]. For a thorough review of synthetic cell–cell communications in both prokaryotes and eukaryotes, see Henning et al. (2015) [34].

While research has elucidated the form and function of cell–cell communication in many biological systems across both unicellular and multicellular organisms, there are often other phenomena at play that confound the effects of communication. For example, the architecture of the genetic network, stochasticity in gene expression, asymmetric cell division, and cell cycle heterogeneity can all contribute to behavioral heterogeneity within a population [35,36,37,38]. In addition to these factors, mechanical interactions, cell death and birth, and environmental conditions can influence the organization of cell types [39,40,41,42]. Furthermore, our knowledge of how communication can impact system behavior is limited by which systems have evolved, which is driven by the functional requirements and historical contingency of each specific system.

Given that cell–cell communication is such a vital aspect across many diverse systems, understanding the extent of its effects can provide widely-applicable insights. Thus, with the work reported here, we strive to push forward the general understanding of what impact cell–cell communication can have on cellular and population behavior. We can apply this understanding to learning how current systems might have evolved, including the transition from unicellular to multicellular life. We can also more fully understand how extant systems operate and how modifications to communication can affect population dynamics. Such knowledge extends to the role of communication in disease-altered states and how disease effects might be remediated by cell–cell communication-based intervention. Lastly, for synthetic systems, it is necessary to understand the role of communication so that systems can be designed accurately and efficiently, and undesirable collateral effects can be avoided.

In order to isolate the effects of communication so that we may understand the general principles relating cell–cell communication to cellular and collective behavior, we use an abstract in silico model of cellular populations. Specifically, we use a 3D multiscale cellular population model, with dynamic intracellular networks interacting via diffusible signals to form intercellular connections (Figure 1). By including intracellular networks, we can more explicitly define cellular behavior and explore the interplay between internally and externally driven behavior. Importantly, cell–cell communication is not a simple present/absent phenomenon in living systems. The mechanism, architecture, and strength of communication are all variable and likely have functional consequences. Therefore, in our work, we focus on two key signaling parameters: the effective interaction distance, which is the distance at which cells are able to interact via signals; and receptor activation threshold, which is the concentration of signal necessary to activate a cell’s receptor. Both of these parameters have previously been shown to tune cells’ relative response to self-generated versus non-self-generated signal [43,44]. Thus, by varying these parameters, which are variable between and within biological systems, we can vary the degree of communication and more finely explore the effects of communication.

We find that cellular populations are characterized by different forms of communication as a function of the effective interaction distance and receptor threshold. These include the disruption of communication altogether and various degrees of dependence between cells within a population. Cellular and population behavior is highly sensitive to both the general form and specific parameters of communication present even when the cellular network has no bias towards that behavior. Tissue diversity, both behavioral and spatial, is also broadly impacted by communication. Overall, this work shows that communication between cells has a strong and tunable effect on biological behavior across a general cellular background that did not evolve a communication system.

## 2. Materials and Methods

### 2.1. Mathematical Model

Our in silico framework consists of a population of individual cells, organized on a 3D grid, which send signals through diffusible molecules (Figure 1a). Each cell is described by an internal Boolean network (BN), consisting of binary-valued regulatory nodes and signaling nodes (Figure 1b). Regulatory nodes, which can be viewed as genes, largely determine the behavior of the cell, while signaling nodes are responsible for signaling between cells. By iterating the state of the system over time, we simulate the dynamics of the cells, and consequently the population. For simplicity, we will refer to a population of cells with identical cellular BNs as a tissue for the rest of the paper. Since a tissue may have different signaling parameters, we will refer to a tissue with specific signaling parameters as a tissue sample.

#### 2.1.1. Cells as Boolean Networks

Each cell in our simulation framework is modeled as a Boolean network. BNs have a long history of being used as biological models, especially as gene regulatory networks [45]. Importantly, BNs are vastly simplified models, as they do not capture many important factors of intracellular dynamics. Here, the reduced detail of BNs is sufficient as our focus is on modeling an abstract cellular system capable of expressing distinct stable behaviors. Moreover, BNs are simple to implement and analyze, and capable of phenomenologically recapitulating many complex behaviors observed in biological cellular systems [46,47].

The BN of cell *i* is a network defined on a set of *n* binary-valued variables,
(1)$$X_{i}=x_{1}^{i},\;\ldots \;,x_{n}^{i}\;,$$
where $x_{j}^{i}\in \left\{0,1\right\}$ represents the expression or activation of node *j*. The nodes in the cellular networks are divided into two categories: regulatory nodes and signaling nodes. Signaling nodes come in pairs of a signal node, which secretes a diffusible molecule, and a receptor node, which is activated by that molecule. Over time, the state of the cell, $X_{i}\left(t\right)$, changes via updating the values of the regulatory and signaling nodes.

The value of each regulatory node, *j*, is determined by applying a corresponding Boolean update function with *k* inputs, $f_{j}^{i}:{\left\{0,1\right\}}^{k}\overset{}{\to }\left\{0,1\right\}$. Thus, the Boolean value of each regulatory node $x_{j}^{i}$ is given by:(2)$$x_{j}^{i}\left(t+1\right)=f_{j}^{i}\left(x_{j_{1}}^{i}\left(t\right),\;x_{j_{2}}^{i}\left(t\right),\;\ldots \;,x_{j_{k}}^{i}\left(t\right)\right).$$

The function $f_{j}^{i}$ can be described by the table of output values (truth table) it gives for each combination of argument values. There are many schemes for generating the architecture of the wiring between nodes (wiring diagram) and truth tables for Boolean networks. For simplicity, we follow the original formulation of the random Boolean network scheme [48]. For the wiring diagram, *k* is constant for all nodes; the *k* input nodes for each regulatory node are randomly chosen uniformly from the regulatory and receptor nodes. The signal node is never used as an input in order to keep communication distinct from internal regulation. For the truth tables, elements are assigned values of 0 or 1 with a bias $p=P\{f_{j}^{i}=1\},j=1,\ldots ,n$, which is the probability that the function takes on the value of 1 for a combination of input values. In closed BNs (i.e., with no external inputs), *k* and *p* can be varied to tune the network’s response to perturbations, ranging between ordered dynamics (insensitive to perturbations) and chaotic dynamics (highly sensitive to perturbations).

The two types of signaling nodes, signals and receptors, each behave differently. The signal node follows the same rules as the regulatory nodes. The receptor node updates using a threshold function on the local concentration of its corresponding signal—described in Section 2.1.3.

#### 2.1.2. Tissue Architecture

In our simulations, each of the *N* cells in a tissue is described by the same BN (identical wiring diagram and truth tables) and occupies a grid point on a 3D structured lattice. Specifically, cells are located in the center of voxels, which have a volume of $V_{c}=H^{3}$ where *H* is the lattice size. We set H=20 for this work with no loss in generality. The lattice has periodic boundary conditions on all three axes, such that all cells have the same neighborhood structure. This allows for observing patterns that might appear in much larger tissues while only simulating a small tissue. It should be noted that simulated tissues are physically static; while the states of the cells change over time, cells do not die, undergo mitosis, or migrate.

#### 2.1.3. Cell–Cell Communication via Diffusible Molecules

We include signaling between cells by modeling the secretion and sensing of a single diffusible molecule, *m*. A cell *i* releases molecule *m* with a secretion rate of $\eta (x_{S}^{i})$ molecules per second, which depends on the Boolean state of its designated signal node, $x_{S}^{i}$. We assume that η(0)=1 and η(1)=α, α>1, to account for basal and active expression, respectively. We make this assumption with no loss of generality since it is equivalent to normalizing active expression by the lower basal expression. We set α=5 in this work. The concentration, *C*, of molecule *m* changes in space and time according to a diffusion degradation equation. For cells arranged in a rectangular lattice, this is approximated by:(3)$$\partial C_{i}/\partial t=D\Delta C_{i}-\gamma C_{i}+\eta \left(x_{S}^{i}\right)/V_{c}$$
for each voxel *i* of the lattice containing cell *i*. *D* is the diffusion coefficient and γ is the constant degradation rate. Assuming that diffusion is much faster than gene regulation, we use the steady state of the diffusion equation above:(4)$$0=D\Delta {\hat{C}}_{i}-\gamma {\hat{C}}_{i}+\eta \left(x_{S}^{i}\right)/V_{c}$$
and use a numerical solver for calculating ${\hat{C}}_{i}$ in simulations. An important component of the steady state solution is the effective interaction distance, λ, where $\lambda =\sqrt{D/\gamma }/R$ and *R* is the radius of the cell [44,49]. λ is scaled by *R* so that the value of λ can be interpreted in terms of multiples of the radius of a cell. When sensing molecule *m*, cell *i* checks the local concentration of the signal at its grid point. If the local concentration of *m* is above a threshold value, θ, then the receptor node is activated; otherwise, it is deactivated. Formally:(5)$$\begin{array}{cc} x_{R}^{i}\left(t+1\right) & =1,\;\mathrm{if}\;{\hat{C}}_{i}\left(x_{S}^{1}\left(t\right),\;...\;,x_{S}^{N}\left(t\right)\right)>\theta , \end{array}$$(6)$$\begin{array}{cc} x_{R}^{i}\left(t+1\right) & =0,\;\mathrm{otherwise}, \end{array}$$
where ${\hat{C}}_{i}\left(x_{S}^{1}\left(t\right),\;...\;,x_{S}^{N}\left(t\right)\right)$ is the concentration of *m* at the position of cell *i*. We will vary λ and θ to change communication within the tissue. The other model parameters are kept constant for all simulations.

#### 2.1.4. Simulation Framework

Simulations of our model were implemented in Biocellion [50], a high performance computing platform designed for simulation of multicellular systems. At every time step *t* of the simulation, the concentration of signalling molecule *m* is updated by numerically solving diffusion Equation (Equation 4), after which the Boolean states of the cells are updated synchronously using the computed concentrations.

## 3. Experiments and Results

In order to investigate the impact of communication on cellular and population behavior, we have taken a generalized in silico approach to study populations of communicating cells. We simulated populations of 4096 cells, organized in a 16 × 16 × 16 3D grid, which send signals within the population through a single diffusible molecular signal. Each population is instantiated with specific signaling parameters λ, the effective interaction distance, and θ, the threshold of the signal molecule’s receptor. By simulating tissues with different signaling parameters, we can probe the effect of communication on cellular and tissue dynamics.

### 3.1. Changing Interaction Distance and Receptor Activation Threshold Can Change the Mode of Communication

In a similar theoretical framework, Maire et al. [44] showed that cells could enter qualitatively different modes of communication depending on signal secretion and receptor threshold. Therefore, we first investigated how the quality of communication that occurs between cells depends on our two signaling parameters λ and θ. Specifically, we tested cells’ ability to influence each other via activation of each other’s receptors.

In this experiment, we constructed a tissue in which the cellular network consists of only the two signaling nodes, with no regulatory nodes (Figure 2a). The update function for the signal node is set to a constant, $x_{S}^{i}\left(t\right)=x_{S}^{i}\left(t=0\right)$, and the receptor node behaves as described in Section 2.1.3. We simulated tissue samples with different λ,θ values to check receptor response (ON,OFF) to distinct combinations of self and neighbor signaling states. λ and θ were sampled using a fine grid of 271 × 141 points in the interval [0.3,3.0] and [1.0,15.0], respectively, and a coarse grid of 66 × 200 points in the interval [0.5,7.0] and [1.0,200.0], respectively. For each λ,θ pair, we initialized a tissue sample with each of four different initial conditions of cell *i*’s and all other cells *j*’s signal nodes, $x_{S}^{i}$ and $x_{S}^{j},j\neq i$ (Figure 2b). We then updated the state of the tissue by a single time step and recorded the response of cell *i*’s receptor, $x_{R}^{i}$. The initial conditions were:All cells’ signal nodes are OFF: $x_{S}^{i}=x_{S}^{j}=0$;Cell *i*’s signal node is OFF and all other cells’ signal nodes are ON: $x_{S}^{i}=0,x_{S}^{j}=1$;Cell *i*’s signal node is ON and all other cells’ signal nodes are OFF: $x_{S}^{i}=1,x_{S}^{j}=0$;All cells’ signal nodes are ON: $x_{S}^{i}=x_{S}^{j}=1$.

By examining whether cell *i*’s receptor node turns ON in each of these scenarios, we are categorizing its self- versus neighbor-dependence. Based on receptor response, we found six distinct signaling regions within our signaling parameter space, which can be split into social (regions S1, S2, and S3) and asocial (regions A1, A2, and A3) categories (Figure 3a). Here, we use the terms social and asocial to indicate whether cells will incorporate the signal originating from other cells in their receptor response. Because of the interplay of λ and θ in receptor activation, the boundaries between these six regions are curves in the λ, θ space. As the interaction distance increases, any cell *i* will receive a greater amount of signal from a greater number of cells, allowing for the activation of its receptor even at higher activation thresholds.

Asocial behavior occurs in regions A1, A2, and A3 of the parameter space (Figure 3a). In region A1, the receptor threshold is so low that a basal level of signal secretion is enough to turn ON a cell’s receptor even when no cells are actively signaling (Figure 3b). As λ increases, more basally produced signal from a wider neighborhood of cells reaches a cell, increasing the maximum threshold that can be satisfied by basal secretion alone. In this region, if any cells did actively secrete signal, the receptor response would be unaffected. We will refer to this region as the ON region. The size of region A1 would decrease if the basal secretion rate was reduced, until it disappears at a rate of 0. In region A3, in contrast, the receptor threshold is so high that even active secretion by all cells will not activate a cell’s receptor. Any changes in the signaling state of the cells would not affect receptor activity. We will refer to this region as the OFF region. Lastly, in region A2, cells can be described as having self-talk behavior. Here, a cell’s receptor is activated only by its own signal, $x_{R}^{i}\left(t+1\right)=x_{S}^{i}\left(t\right)$ (Figure 3b). Even when receiving the maximum amount of neighbor signal, a cell’s receptor remains OFF unless the cell itself has signaling turned ON. This self-only activation can be due to either a cell’s signal not reaching far enough to affect neighboring cells or a cell’s own signal masking that of its neighbors. We will refer to this region as the SELF region. Thus, in regions A1, A2, and A3, cells are behaving completely independently of each other, with receptor response constant under any configuration of neighbor signaling states.

Signaling becomes communication and cells’ receptor responses are affected by their social environment in regions S1, S2, and S3 of the λ, θ space (Figure 3a). In the extreme case, in region S2, the state of a cell’s receptor is largely determined by the signal secreted by its neighbors (Figure 3b). A cell cannot activate its own receptor but can contribute to its receptor activation in concert with signaling neighbors. Neighbor signal, on the other hand, can activate a cell’s receptor on its own. In parameter regions S1 and S3, a cell’s receptor is more equally affected by both self and neighbor signal. Specifically, in region S1, the receptor can be activated by *either* itself or neighbors; in region S3, a cell must receive signal from *both* itself and neighbors for its receptor to be activated. For each of these regions, the configuration of signaling states within the population affects receptor response and more neighbor signal is required as θ is increased.

While these results were obtained from experimental simulations, one can also analytically derive the four boundary curves necessary to define the regions in the signaling parameter space using an approximation of the steady state equation for molecule concentration (see Appendix B). This analysis also reveals that the relative positioning of the boundary curves is independent of many biological parameters, including: the basal secretion rate, the active secretion rate, the diffusion coefficient, the radius of cells, the spacing between cells, and the size of the population. Thus, as long as these parameters are constant within a population, the relative distribution of the regions and their boundaries will not change.

Here, we have only considered the extreme case in which every neighbor cell is either signaling or not. However, the switch in social behavior occurs exactly at these extremes. Within each parameter region, the particular response of a cell is dependent on the fraction of signaling neighbors, their spatial configuration, and the signaling parameter values; nevertheless, the particular response will fall under the category determined by the parameter region. The granular effects of signaling state configuration were explored by Maire et al. [44] in a similar theoretical framework. In examining the role of the secretion rate and receptor activation threshold in determining the relative autonomy of bacteria, they demonstrate the same social dynamics as we have.

### 3.2. Cellular and Tissue Behavior Are Distinct between Asocial and Social Behavior

Having identified six distinct signaling regions, we then investigated how cellular and tissue behavior can change between and within them. For this, we simulated tissues with different λ, θ values and analyzed the resulting cellular and tissue dynamics. In this experiment, we generated 100 unique tissues, each with a particular cellular Boolean network. The cellular BNs consist of 12 nodes (10 regulatory, 1 signaling pair), allowing for a variety of possible cellular dynamics. Each cellular BN is instantiated as described in Section 2.1.1, using in-degree k=3 and bias p=0.2113. These values were chosen to give dynamically critical cells, the proposed dynamics of living cells [51,52]. Though it should be noted that artificial cells that do not communicate are likely to be critical at these parameter values; however, when communication is introduced, the dynamical regime of the cells may be altered and may no longer be critical.

We simulated each tissue with 108 λ, θ parameter values (12 λ × 9 θ values) that were sampled from across the different signaling regions identified in the previous experiment. The 12 λ values are incremented by a value of 0.5, equivalent to half the cell radius, and the 9 θ values are sampled such that the different signaling regions are represented—1 value in A1, 1 in S1, 5 uniformly in S2 or A2, 1 in S3, and 1 in A3 (Appendix A). The initial states of all cells in a tissue were chosen randomly, with the same initial conditions for tissue samples of the same tissue. Thus, any variations in dynamics are attributable to changes in communication only.

All simulations were run for 2000 time steps, at which point all tissue samples had reached steady state behavior (attractor). A tissue sample reaches an attractor once all cells collectively revisit the same cell states, i.e., the set of all nodes in the tissue sample revisits the same state. Formally, this occurs when there exists a τ>0 such that:(7)$$X_{i}\left(t\right)=X_{i}\left(t-\tau \right),\;i=1,...,N.$$

The tissue attractor is defined as the vector of all cell states that occurred between time t−τ and *t*:(8)$$A^{T_{\lambda ,\theta }}=\left[A^{C_{1}},\;A^{C_{2}},\;...\;\;A^{C_{N}}\right]$$
where
(9)$$A^{C_{i}}=X_{i}\left(t-\tau \right),\;X_{i}\left(t-\tau +1\right),\;...\;,X_{i}\left(t-1\right),\;i=1,\;...\;,N.$$
is the cellular attractor (CA) occupied by cell *i*. Note that, typically, attractors are fully ordered sets under circular permutation; however, we do not consider state order. In practice, we have observed that the state order is never the sole discerning feature between two cellular attractors.

For each tissue, there exists a set of unique CAs, $A_{1}$, $A_{2}$, …, $A_{M}$ such that, for every i∈{1,…,N}∃j∈{1,…,M} such that $A^{C_{i}}=A_{j}$. For analysis, we identified and characterized the attractors of the cells and tissues.

#### 3.2.1. Cellular Behavior

First, we simply considered whether a cell behaves differently when cell–cell communication is present (S parameter regions) versus absent (A parameter regions). To quantify this question, we measured the likelihood that a cell’s behavior would change as a function of the signaling parameters. For a given tissue sample, we identified which cellular attractor each of its 4096 cells occupied, representing the tissue sample by its tissue attractor, $A^{T_{\lambda ,\theta }}$. The likelihood of a cell to change its behavior was measured as the fraction of cells in a tissue sample that are in a different cellular attractor compared to when no communication occurs. We use tissue samples in the three asocial regions (ON, OFF, and SELF) as references for changes in cellular behavior. To compare a social tissue sample to each asocial reference, we use the normalized Hamming distance:(10)$$D_{H}\left(A^{T_{\lambda ,\theta }},A^{T_{\lambda^{\prime },\theta^{\prime }}}\right)=\frac{\sum_{i=1}^{N}\chi \left(A^{T_{\lambda ,\theta }}\left(i\right)=A^{T_{\lambda^{\prime },\theta^{\prime }}}\left(i\right)\right)}{N}$$
where $A^{T_{\lambda^{\prime },\theta^{\prime }}}$ is one of $A^{T_{\lambda^{ON},\theta^{ON}}}$, $A^{T_{\lambda^{OFF},\theta^{OFF}}}$, or $A^{T_{\lambda^{SELF},\theta^{SELF}}}$—the CA sets corresponding to the three asocial regions. χ(a) is an indicator function that outputs a one if *a* is true and zero if *a* is false. $D_{H}$ is thus a fraction ranging between 0 and 1. We use the minimum of these three values:(11)$$b_{\lambda ,\theta }=min\left(D_{H}\left(A^{T_{\lambda ,\theta }},A^{T_{\lambda^{ON},\theta^{ON}}}\right),D_{H}\left(A^{T_{\lambda ,\theta }},A^{T_{\lambda^{OFF},\theta^{OFF}}}\right),D_{H}\left(A^{T_{\lambda ,\theta }},A^{T_{\lambda^{SELF},\theta^{SELF}}}\right)\right)$$
as our metric of cellular change so that we measure the changes as compared to the most similar asocial reference. We find that $b_{\lambda ,\theta }$ takes the full range of values, with the frequency decreasing as the likelihood of cellular change increases (Figure 4a). Surprisingly, we find that about one third of tissues (36%) are not affected by communication at all, which is greater than that expected by chance due to the regulatory nodes being disconnected from the receptor node (5%). In fact, analysis of the 100 BNs showed that only 8% were disconnected in such a way. Despite the presence of receptor-regulatory wiring in the other 28% of tissues, cells do not change their CA regardless of how communication is tuned (Figure 4b). In all but one of these tissues, a cell’s behavior in the social parameter regions is identical to that in the SELF region. Moreover, the SELF region in these tissues is identical to that in either the ON or OFF parameter regions, indicating that the signal molecule is either always secreted or never secreted (not shown). In all these instances, cells maintain asocial behavior even when communication occurs within the tissue sample.

In the other 64% of tissues, some fraction of cells in the social tissue samples entered different CAs compared to the ones occupied in the asocial references. Here, we found that the magnitude of the cells affected is not only different between tissues but within tissues (across tissue samples) as well (Figure 4c–e). Thus, the fraction of cells with altered behavior is highly dependent on the value of λ and θ. We observed three general trends in $b_{\lambda ,\theta }$ for a tissue. Within the social regions, $b_{\lambda ,\theta }$ can be equal across all λ, θ values (Figure 4c), have a single peak that roughly follows a curve in λ, θ space (Figure 4d), or have two peaks that roughly follow curves in λ, θ space (Figure 4e). Additionally, these tissues can behave identically to the ON, OFF, and SELF regions even when the effective interaction distance and receptor activation threshold place the tissue in the social regions (Appendix A). For each of these general trends, the particular response of tissue samples across λ,θ space is variable between tissues (Appendix A). Interestingly, the greatest increase in $b_{\lambda ,\theta }$ occurs within a short neighborhood of communication (λ = 3) (Figure 5a). Overall, it is clear that changing the quality of communication between cells can lead to measurable changes in a cell’s behavior, with significant diversity in the responses of individual tissues.

For those tissues in which cellular behavior is affected by cell–cell communication, we also considered whether it can alter the range of possible behaviors cells can exhibit. In other words, is cellular behavior limited or expanded by communication? For each tissue, we counted the number of all cellular attractors observed across either asocial tissue samples (regions A1, A2, A3) or social tissue samples (regions S1, S2, S3) and found a significant difference between the two (Figure 6a). In roughly 50% of both social or asocial tissue groups, cells were observed in five or fewer CAs; however, the tail of the distribution for social tissues was heavier and longer than that for asocial cells (Figure 6b).

Given the discrepancy between the number of cellular attractors available to cells in social versus in asocial tissue samples, the set of CAs in each category must be distinct, i.e., cells that communicate are capable of different behavior compared to cells that do not. By directly comparing CAs between the asocial and social parameter regions, we found that cellular BNs can both lose or gain CAs when communicating (Figure 7). A cellular BN loses a CA when communicating if that CA is present in asocial tissue samples but absent in social tissue samples. Inversely, a CA is gained if it is absent in asocial tissue samples but present in at least one social tissue sample. Losing or gaining behaviors is not rare, with 66% cellular BNs missing at least one CA (Figure 7a) and 34% cellular BNs gaining at least one CA (Figure 7b), with some cellular BNs both missing and gaining CAs. Moreover, the behaviors that are either gained or lost are expressed in a nontrivial number of cells (Appendix A), and could therefore be biologically meaningful.

While missing CAs are missing from the entire social signaling parameter space, the expression of novel CAs is not evenly distributed across interaction distances and receptor threshold values. Some novel CAs are only observed at particular λ, θ values. We found that, on average, a cellular BN expresses the largest fraction of its novel CAs at λ=1.5 (Figure 5c). Additionally, the maximum fraction of cells in a novel CA within a tissue sample increases with λ until λ=3 (Figure 5d). Thus, a cell will have the highest chance of expressing a novel behavior and the largest number of possible novel behaviors, when communication is limited to the distance of one or two neighboring cells.

In order to understand how novel CAs can arise in cells in social tissue samples, we considered how they are related to other CAs. Per the definition of a cellular attractor, a CA consists of a set of cellular states. In the case of cells in asocial tissue samples, when cells are essentially closed systems, a cell state *X* will only occur once and will always have the same following cell state $X^{\prime }$. Therefore, the CAs of cells within a particular asocial parameter region are composed of completely disjoint sets of states. When communication is introduced, cells become open systems, and the composition of their CAs is no longer constrained. Thus, it is possible that the novel CAs are variations of existing CAs. To obtain an idea of how CAs that arise in social tissue samples are related to those in asocial tissue samples, we used the set intersection between CAs to calculate the overlap between each novel CA and each asocial CA. Using this method, novel CAs can be classified into three broad categories: modified, combined, and true novel. A modified CA is one that has a non-empty set intersection with one (and only one) asocial CA with the addition of other cell states. These additional cell states are transient states in asocial cells that become stabilized in social cells. A combined CA is one that has a non-empty set intersection with more than one asocial CA with or without the addition of cell states. The final category, true novel, describes a CA that has an empty set intersection, i.e., no shared states, with all asocial CAs. These true novel CAs are composed entirely of states that were unstable transients in asocial cells. Notably, a cellular BN can have one, two, or all three types of novel CAs. Overall, we found that combined CAs are the most common category, being present in the highest fraction of tissues (Figure 8a) and usually the highest fraction of the novel CAs for any of one tissue (Figure 8b). Modified CAs are the next most common followed by true novel.

#### 3.2.2. Tissue Behavior


**Does the composition of a tissue change as communication is added?**


Next, we examined the relationship between communication and population behavior. Since changes in lower cellular-level properties are not perfectly correlated with changes in higher tissue-level properties, we began by testing whether a tissue has any change to its behavior in response to changes in communication. Specifically, we measured the macroscopic property of CA distributions within the tissue samples. To construct a CA distribution for a given tissue sample, we calculate the fraction of cells in the tissue with each CA, which gives:(12)$$D_{\lambda ,\theta }=d_{\lambda ,\theta }^{1},d_{\lambda ,\theta }^{2},\;...\;,d_{\lambda ,\theta }^{M}\;,$$
where
(13)$$d_{\lambda ,\theta }^{i}=\frac{\sum_{j=1}^{N}\chi \left(CA_{j}=A_{i}\right)}{N}$$
and *M* is the number of unique CAs for that tissue across all interaction distances, λ, and receptor thresholds, θ. Each $D_{\lambda ,\theta }$ for a given tissue has the same number of elements, *M*, regardless of the CAs observed at a particular value of λ,θ. To measure changes to a CA distribution, we compare distributions between tissue samples using the symmetric Kullback–Leibler (KL) divergence. The symmetric KL divergence for *P* and *Q* is given by:(14)$$D_{KL}\left(P,Q\right)=D_{KL}\left(P||Q\right)+D_{KL}\left(Q||P\right)$$
which is the sum of the two KL divergences:(15)$$D_{KL}\left(P||Q\right)=-\sum_{x\in X}P\left(x\right)\log\left(\frac{Q(x)}{P(x)}\right)$$
and, similarly defined, $D_{KL}\left(Q||P\right)$. Here, we use the natural logarithm, though any base may be used for the logarithm. If *P* and *Q* are identical distributions, then the symmetric KL divergence will be 0 and will increase as *P* and *Q* diverge. In practice, a small value ϵ<<1 is added to *P* and *Q* to avoid calculations involving a value of 0.

In our calculations, the two distributions (*P* and *Q*) that we compare are the CA distribution of a tissue sample, $D_{\lambda ,\theta }$, and the CA distribution of a tissue sample in one of the three asocial regions: $D_{\lambda^{ON},\theta^{ON}}$, $D_{\lambda^{OFF},\theta^{OFF}}$, $D_{\lambda^{SELF},\theta^{SELF}}$. As before, we are using the asocial regions as references to quantify the effect of changing communication, taking the minimum across the three references. Therefore, we define:(16)$${\hat{D}}_{KL}\left(\lambda ,\theta \right)=min\left(D_{KL}\left(D_{\lambda ,\theta },D_{\lambda^{ON},\theta^{ON}}\right),D_{KL}\left(D_{\lambda ,\theta },D_{\lambda^{OFF},\theta^{OFF}}\right),D_{KL}\left(D_{\lambda ,\theta },D_{\lambda^{SELF},\theta^{SELF}}\right)\right)$$
as our metric of change in tissue composition.

We found that changes in tissue composition range between 0 and 72, approximately the full range of values (Figure 4f). Lower values are more frequently observed. As expected, the 36% of tissues that showed no change in cellular behavior also show no change to tissue composition (Figure 4g). For the other 64% of tissues, there is a mixed relationship between the likelihood of cellular change and changes in tissue composition. In some tissues, there is a clear positive correspondence (Appendix A). In others, there seems to be little change in tissue composition regardless of changes in cellular behavior (Appendix A). Most interestingly, some tissues have little change in tissue composition until some percentage of cells switch behavior (Appendix A).

Overall, tissue composition is clearly a function of the signaling parameters (Figure 4h–j). The broad regions of asocial and social behavior in the parameter space are reflected in the shifts of tissue composition. Even so, the response of each tissue is unique, with diversity across and within tissues. The same three general trends observed for changes in cellular behavior (flat, single peak, double peak) are also observed for changes in tissue composition. In many cases, the tissue composition of the ON, OFF, or SELF regions is observed in the social regions, minimizing the communication parameter space where distinct tissue compositions can occur (Appendix A). Looking across all of the tissues, we find that, on average, tissue composition changes most rapidly as the effective interaction distance increases beyond a single cell, peaking by λ=3 and then leveling out (Figure 5b).


**Does the diversity of a tissue depend on communication?**


Taking a different approach to measuring tissue composition, we also analyzed the diversity within tissues. Population diversity is a functionally important property of cellular populations, and is tightly regulated, whether for generating homogeneity or heterogeneity [3,9,21]. Using the CA distributions of tissue samples, we calculated the entropy of the distribution as a measure of diversity—completely homogeneous tissue samples having the lowest entropy and tissue samples with an even distribution of every CA having the highest entropy. Tissue diversity is defined as:(17)$$H_{\lambda ,\theta }=-\sum_{i=1}^{M}d_{\lambda ,\theta }^{i}\;\log\left(d_{\lambda ,\theta }^{i}\right).$$ First, we examined the specific case of homogeneous tissue samples, in which $H_{\lambda ,\theta }=0$. We found a clear trend between the fraction of tissue samples that are homogeneous and the degree of communication (Figure 9a). At λ=1, there is an immediate decrease in the number of tissues with only one cell type observed at that λ. This is the λ value at which social communication begins in our parameter space. Remarkably, it is also the global minimum of the fraction of homogeneous tissues. As the effective interaction distance and receptor activation threshold are increased along the curve, the fraction of homogeneous tissues increases. Notably, this fraction is both highest and lowest within the social regions, but peaks or dips at a low λ value (Figure 9b).

For heterogeneous tissue samples, we found that communication has a strong effect on tissue diversity. While the tissue samples in the asocial and social regions have a similar distribution of entropy values, social tissue samples exhibit more entropy values per tissue and a 60% increase in the maximum entropy as compared to asocial tissue samples (Figure 4k). Tissue entropy in social tissue samples take values that are higher, lower, and/or the same as in asocial regions, varying along λ, θ curves (Figure 4n,o).

We also briefly investigated the spatial heterogeneity in the tissues by visualizing the cells in the tissue samples as cubes colored by a cellular attractor. While many tissues did not have qualitatively notable spatial patterning (not shown), some displayed distinctive patterning (Appendix A) resembling the reaction–diffusion patterns observed in biological tissues [53]. This spatial patterning was highly dependent on the signaling parameters.

## 4. Discussion

Communication is a key component of living systems, providing a means by which cells can coordinate their activities to achieve collective goals and accomplish complex functions. Understanding the range and magnitude of the effects of communication within populations advances our understanding of how communication systems have evolved and how they function in extant organisms. It also advances our ability to modify, manipulate, and repair existing or synthetic systems. In this work, we have used an in silico model of three-dimensional cellular populations to explore the effects communication can have on cellular and population behavior. In order to capture the interplay between intercellular communication and intracellular systems capable of diverse behavior, we modeled cellular dynamics using random Boolean networks. Boolean networks have been successfully utilized to capture cellular dynamics with multiple stable behaviors, including the transition into or between behaviors upon external perturbation [46,47,54,55,56]. However, BNs are limited as mechanistic models of the hierarchical differentiation that occurs in multicellular organisms. Many key features of gene regulation, such as epigenetic regulation, enhancer-based transcription, and non-switch like reactions, are not captured by the BN formalism and thus limit the behaviors that it can model [57]. Though we do not focus on demonstrating the full extent of differentiation here, these limitations should be noted as caveats to the application of our results to specific eukaryotic systems. Overall, by modeling an abstracted system, we aimed to explore the generalized effects of communication without focusing on the details of cellular biology and without the bias of systems evolved for specific functions.

We focused on two signaling parameters, the effective interaction distance between cells, λ, and the receptor activation threshold, θ. These parameters have previously been shown to affect the degree of communication between cells [43,44]. In addition, both parameters are variable within and between populations [58,59,60]. The effective interaction distance between cells can be altered by the biochemical properties of the signaling molecule, binding properties of the signal receptor, and the contents of the extracellular environment, all of which can influence the stability, concentration, and diffusion rate of the signal molecule. The receptor activation threshold can be altered by the biochemical properties of the receptor and the level of receptor expression by the cell, both of which can change the likelihood of receptor activation in the presence of signaling molecule.

Using our model, we have shown that signaling by cells does not guarantee communication, in which a signal sent by one cell can alter the receptor response of another (Figure 3a). Moreover, depending on λ and θ, cells can exhibit very different social dependencies. They can be completely asocial, either with receptors always ON/OFF or solely dependent on self-generated signal. Alternatively, they can be social, either with receptor activation only dependent on neighbor-generated signal or on some combination of self- and neighbor-generated signal. Similar signaling categories have been shown by Maire et al. [44] who explored the role of the secretion rate and receptor activation threshold in a similar framework. Thus, changing λ and θ are possible mechanisms by which populations that use the same signaling framework can exhibit different levels of social dependency.

When biological cells do communicate, they coordinate by regulating cellular behavior within the population. Thus, cells utilize external signals as cues for changing their behavior. We found that the fraction of cells within our simulated populations that change behavior due to communication is highly variable with λ and θ (Figure 4a). It is decreasingly likely to observe larger fractions of cellular change, with many tissues showing no changes at all. This demonstrates the relative influence of the internal dynamics of the cell compared to the external social influence of the population. Thus, both the signaling parameters and internal cellular networks must meet certain criteria for social interdependence to occur between cells. Considering that the single receptor node is competing with the influence of multiple regulatory nodes to determine cellular dynamics, it is reasonable that stronger receptor influence on cell behavior would be increasingly rare.

One consequence of the external influence from other cells is that the set of behaviors that cells can exhibit is not consistent between cells within asocial and social signaling environments. Cellular behaviors can be lost or gained due to cell–cell communication (Figure 7a,b). Interestingly, Damiani et al. [61] showed that new behaviors can also arise in communicating populations when changing coupling strength, i.e., the number of unique signals used in communication. Fundamentally, these findings mean that asocial and social cellular populations are operating within distinct sets of functionalities. That is, cells can not only use cell–cell communication to coordinate the expression of different cellular behaviors, but to define what behaviors are even possible within asocial or social settings. One repercussion of this is that disrupting communication in social populations can either result in the removal of social-specific behaviors or the appearance of asocial-specific behaviors. Similarly, introducing communication to an asocial population can result in the loss of certain behaviors or the appearance of unpredicted behaviors. Further examination of how distinct cellular behaviors can arise in social populations showed that they can either be variants of asocial behaviors, combinations of asocial behaviors, or completely distinct from asocial behaviors (Figure 8a). Since cellular behaviors are often linked to specific functions, these categories may have functional significance. Specifically, social behaviors that are variations or combinations of asocial behaviors may be functionally similar and social behaviors that are distinct may have distinct functions. By generating these novel behaviors, cell–cell communication could be producing alternative cellular functions rather than simply triggering them to occur. Notably, since our work involved randomly generated cellular networks, our results show that the loss and gain of cellular behaviors is an emergent property of populations in which cells communicate rather than a product of evolutionary fine-tuning.

While the behavior of individual cells is regulated by intercellular signals, cell–cell communication is ultimately a population phenomenon. As such, population-level properties are also regulated by communication as a culmination of cellular level changes. In this work, we focused on the distribution and diversity of cellular behaviors within a tissue (tissue composition, tissue diversity) as population properties that are often important in achieving collective behavior, especially division of labor. We have shown that both tissue composition and diversity can be tuned by changing the effective interaction distance and receptor activation threshold (Figure 4f,k). Interestingly, though changes in the behavior of individual cells is a requirement for changes in tissue composition, the relationship between the two is not the same for all tissues (Appendix A). Consequently, cellular populations can have separate control over coordinating cellular behavior and changing the overall makeup of the population, such as in the case of spatial organization. It may not be functionally desirable to change which cellular behaviors are expressed, but simply where. We also showed that, when communication occurs, the percent of behaviorally homogeneous tissues increases with interaction distance (Figure 9b). Relatedly, Damiani et al. [61] showed that the percent of homogeneous tissues increases with the coupling strength. These findings together indicate that a stronger connection between cells, whether it be through greater coupling to one cell or more communication partners, can strengthen the synchronization between them. In complete contrast to this, however, we also observed tissues in which cell–cell communication leads to an increase in behavioral diversity, demonstrating that communication can equally lead to unifying or diversifying coordination. In terms of spatial rather than behavioral diversity, we showed that spatial patterning similar to that observed in reaction–diffusion systems occurs in our tissue models in a communication-dependent manner (Appendix A). However, we did not observe the regular patterns often found in biological systems, such as spots or stripes. Our sampling of network rules and initial conditions may have been too limited to observe these behaviors by random chance. Similarly, without sampling more initial conditions under the same signaling parameters, we do not know whether the patterns we did observe are robustly expressed within tissues. More targeted experiments and quantitative analyses are necessary to further investigate spatial patterning in this model.

For each of the cellular and tissue properties we have discussed, we have found that not only do different tissues have different responses to changing cell–cell communication parameters, but that a single tissue is capable of distinct responses within the social regions of the λ, θ signaling parameter space. However, there is one overarching trend that can be seen across both cellular and population metrics. As the neighborhood of interaction is extended beyond two neighboring cell lengths, there is a diminishing response to cell–cell communication. On average, both the sharpest change and greatest magnitude of response is observed within this range. Effectively, populations of cells do not need to interact beyond a range of one or two cells to utilize the range of effects we have demonstrated here. Since the same signal is generated by all neighbors, regardless of distance, it is reasonable that the quality of cellular and population behavior would not depend on the distance the signal has traveled. This quality may extend to other properties resulting from cell–cell communication.

Overall, there is a clear difference between the cellular and population behavior of communicating and non-communicating populations. Due to the small sample size of tissue BNs that were simulated (100 of $10^{10}$ Boolean networks), more simulations would be necessary to fully demonstrate the robustness of this observation. However, genetically diverse populations all exhibit the same set of trends in their response to cell–cell communication, indicating that we have captured nontrivial behavior. Within the general trends, there is still flexibility in the specific responses of different populations across cellular and population properties, though the factors responsible for this diversity are unclear. It may be that the tissues are inherently capable of different responses due to network structure and dynamics. Alternatively, it may be that all tissues are capable of similar responses, but we have only observed the outcome of a single initial condition. Further simulations and network characterization are necessary to explore these possibilities. For any particular tissue, cellular populations can switch between broad types of asocial and social behavior by tuning their signaling parameters. Even within social populations, there is heterogeneity and flexibility in response to different values of the signaling parameters. Furthermore, in real cellular populations, the properties of signaling molecules can vary across space and time. Different signaling molecules will also have their own properties. Thus, social behavior within a single population is spatially and temporally dynamic and functionally modular. Additionally, external factors may modify signaling parameters, allowing for cells to utilize environmental cues to alter their behavior or for an external agent to manipulate the system. This diversity enables specialization for the unique selective pressures of any given population, tuning their signaling parameters over both the lifetime of a single cell and evolutionary time.

## Figures and Tables

**Figure 1 entropy-25-00319-f001:**
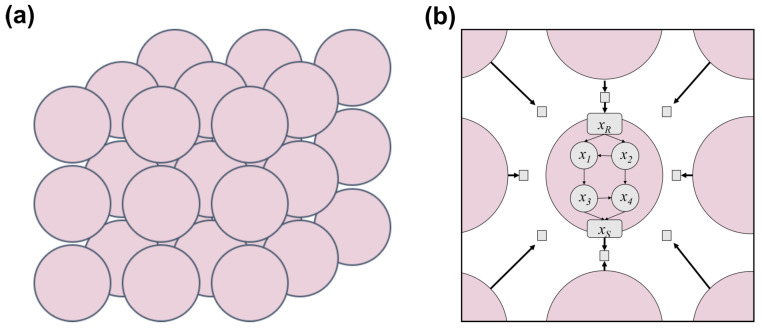
3D structured population of communicating cells. (**a**) The cellular population model consists of cells (pink circles) that are arranged in three dimensions on a lattice-structured grid. (**b**) Cells are described by internal networks and communicate with each other by secreting and sensing diffusible molecules (gray squares).

**Figure 2 entropy-25-00319-f002:**
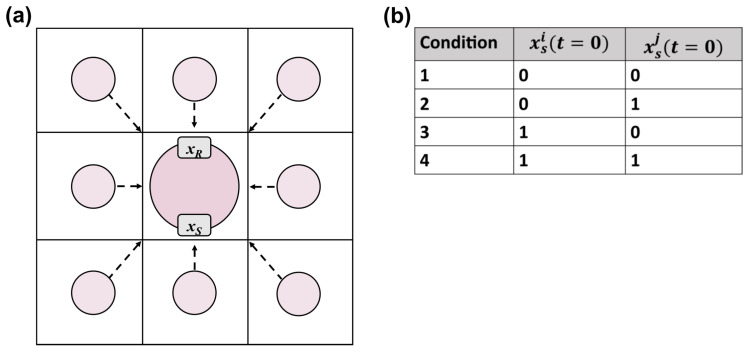
Experimental design for testing the asocial to social transition of cells. (**a**) a population of simple cells with only 2 nodes, $x_{S}$ and $x_{R}$; (**b**) the initial conditions for the simple 2-node experiment. $x_{S}^{i}\left(t=0\right)$ is the initial state of cell *i*’s signaling node, and $x_{S}^{j}\left(t=0\right)$ is the initial state of all other cells’ *j* signaling nodes.

**Figure 3 entropy-25-00319-f003:**
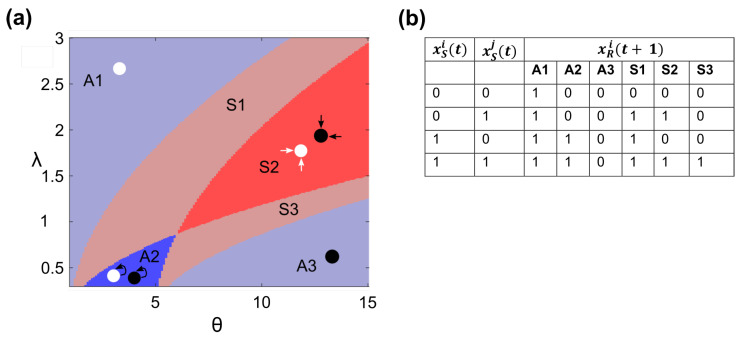
(**a**) The six distinct signaling regions in the λ, θ space. Each region corresponds to a distinct signaling behavior of any cell *i* in a tissue sample with a given λ, θ value. In the asocial regions (blue), cell *i*’s receptor is always ON (A1—light blue), responds to self-signal (A2—dark blue), or is always OFF (A3—light blue). In the social regions (red), cell *i*’s receptor can be activated by self or neighbor signal (S1—light red), primarily neighbor signal (S2—dark red), or both self and neighbor signal (S3—light red). For easier visualization, only λ≤3 is shown. See Appendix A for the wider parameter range used in simulations; (**b**) the value of cell *i*’s receptor at time t=1, $x_{R}^{i}\left(t=1\right)$, in response to the four initial signaling conditions of cell *i*, $x_{S}^{i}$, and all other cells *j*, $x_{S}^{j}$, at time t=0.

**Figure 4 entropy-25-00319-f004:**
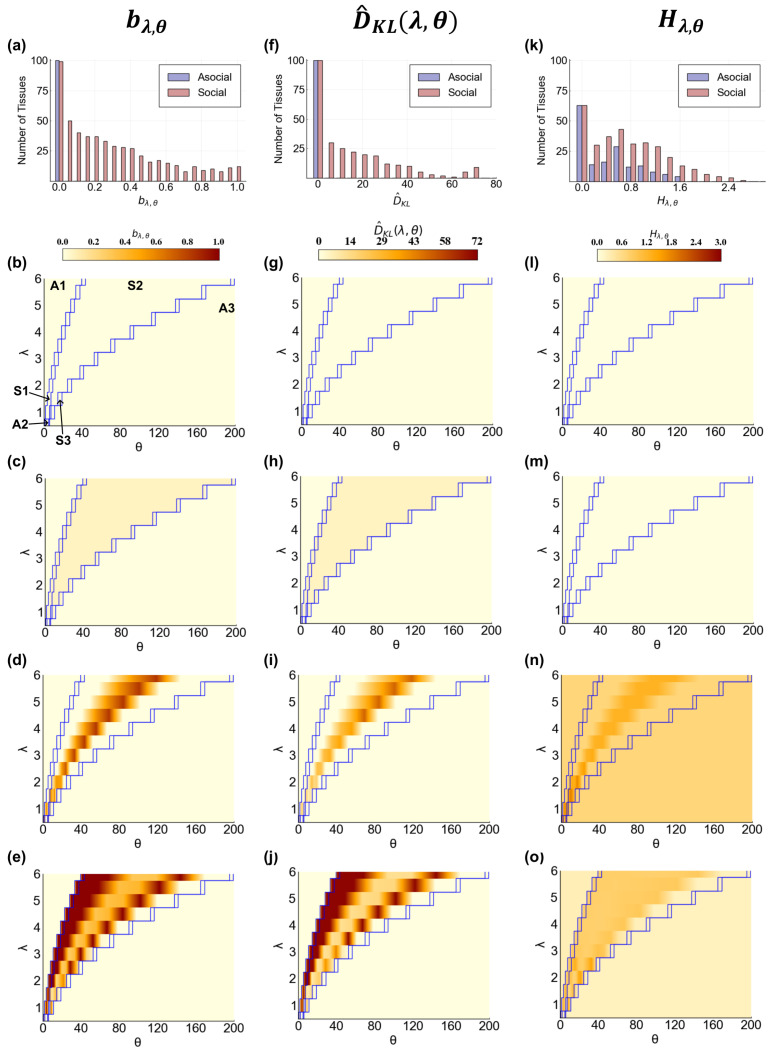
The effects of communication on cellular and population behavior. (**a**) the distribution of $b_{\lambda ,\theta }$ for all 100 tissues divided between the asocial(light blue) and social (light red) λ, θ regions. Each bar counts the number of (a)social tissues with at least one $b_{\lambda ,\theta }$ at that value. All $b_{\lambda ,\theta }=0$ for asocial tissues by definition; (**b–e**) $b_{\lambda ,\theta }$ (color) for four example tissues across all sampled λ,θ values; (**b**) no changes in cellular behavior; (**c**) no variation in $b_{\lambda ,\theta }$ within social regions; (**d**) a single peak of high $b_{\lambda ,\theta }$; (**e**) two peaks in $b_{\lambda ,\theta }$; (**f**) the distribution of ${\hat{D}}_{KL}$ for all 100 tissues as in (**a**); (**g–j**) ${\hat{D}}_{KL}$ for four example tissues across all sampled λ,θ values; (**g**) no change in ${\hat{D}}_{KL}$; (**h**) no variation in ${\hat{D}}_{KL}$ in social regions; (**i**) a single peak in ${\hat{D}}_{KL}$; (**j**) two peaks in ${\hat{D}}_{KL}$; (**k**) the distribution of $H_{\lambda ,\theta }$ for all 100 tissues as in (**a**); (**l–o**) $H_{\lambda ,\theta }$ for four example tissues across all sampled λ,θ values; (**l**) no change in $H_{\lambda ,\theta }$; (**m**) no variation in $H_{\lambda ,\theta }$ despite changes in cell and tissue composition; (**n**) a single peak in $H_{\lambda ,\theta }$; (**o**) two peaks in $H_{\lambda ,\theta }$. For all heatmaps, each row represents the same tissue across different metrics. Blue lines represent the four boundaries of communication regions as measured in Section 3.1. Data between sampled θ values for each λ value are interpolated for better visualization.

**Figure 5 entropy-25-00319-f005:**
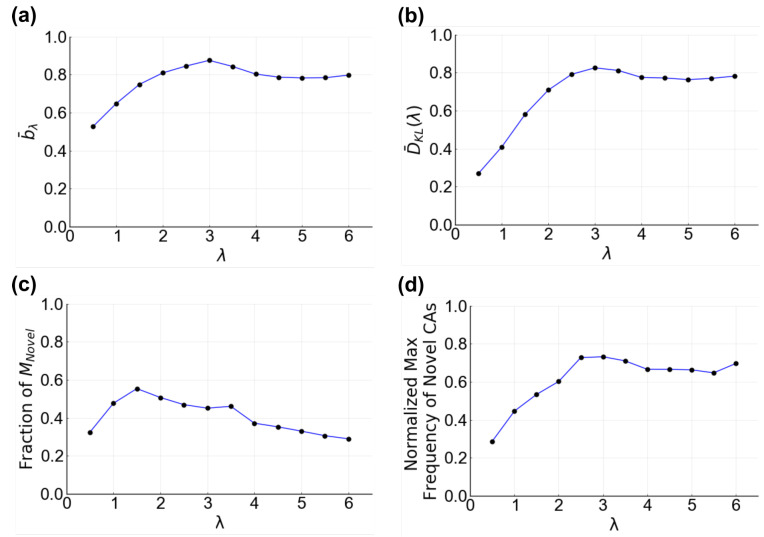
Metrics of cellular change as a function of λ. (**a**) normalized maximum $b_{\lambda ,\theta }$ as a function of λ, ${\bar{b}}_{\lambda }$. For each λ value, the maximum $b_{\lambda ,\theta }$ value is calculated and normalized by the maximum of the tissue. The mean of the maxima across all 100 tissues is plotted; (**b**) ${\hat{D}}_{KL}$ as a function of λ, ${\bar{D}}_{KL}\left(\lambda \right)$. For a given λ value, the maximum ${\hat{D}}_{KL}$ was calculated and normalized for each tissue. The mean across all tissues is plotted; (**c**) observation of novel CAs as a function of λ. $M_{Novel}$ is the number of unique Novel CAs for a given tissue. The number of Novel CAs expressed at each λ is calculated as a fraction of $M_{Novel}$ and the mean across all tissues is plotted; (**d**) frequency of Novel CAs as a function of λ. The maximum fraction of cells expressing a novel CA is calculated for each λ and normalized by the tissue maximum. The mean of the normalized maximum fraction (black circles) is plotted with a trend line (blue line).

**Figure 6 entropy-25-00319-f006:**
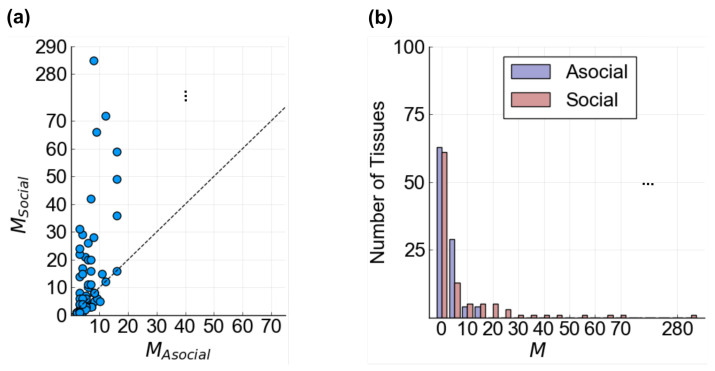
Numbers of cellular attractors observed in asocial (A1, A2, A3) and social (S1, S2, S3) regions of the λ,θ parameter space. $M_{Asocial}$ and $M_{Social}$ are the number of unique CAs observed in the asocial and social λ,θ regions, respectively. (**a**) $M_{Asocial}$ vs. $M_{Social}$. Each tissue’s $M_{Asocial}$ is directly plotted against its $M_{Social}$. The y-axis has been condensed (triple dots) to better visualize outlying data points. The diagonal line is plotted for reference (dashed line). It is rare that a tissue would have fewer attractors observed in social conditions as compared to asocial conditions (points below the diagonal); (**b**) histograms of $M_{Asocial}$ and $M_{Social}$. The x-axis has been condensed (triple dots) to better visualize outlying data points. Social tissues exhibit a longer tailed distribution as compared to asocial tissues, with a maximum of over 10 times as many cellular behaviors.

**Figure 7 entropy-25-00319-f007:**
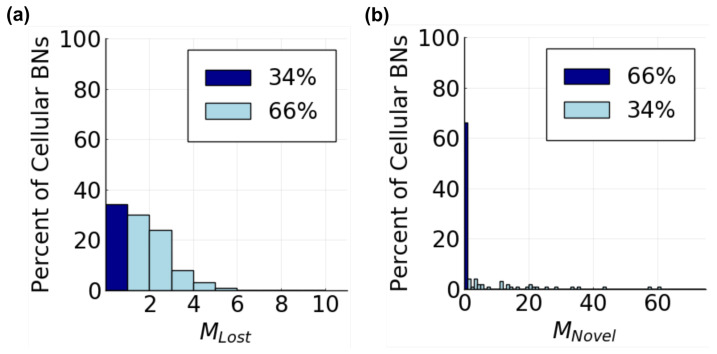
Likelihood of observing Lost or Novel CAs in tissues. (**a**) distribution of $M_{Lost}$, the number of CAs lost by each tissue when communicating. In addition, 66% of cellular BNs lose at least one CA (light blue) and 34% do not lose any CAs (dark blue); (**b**) distribution of $M_{Novel}$, the number of Novel CAs gained by each tissue when communicating. Furthermore, 34% of cellular BNs gain at least one CA (light blue) and 66% do not gain any CAs (dark blue).

**Figure 8 entropy-25-00319-f008:**
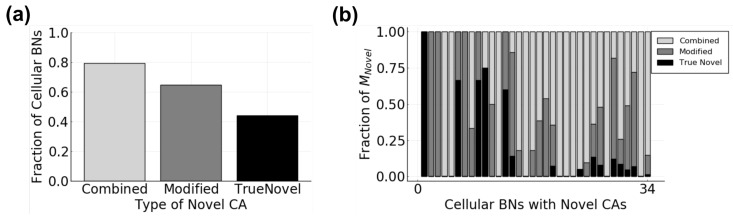
Characterization of Novel CAs. (**a**) fraction of cellular BNs with each of the three categories of Novel CAs; (**b**) distribution of each category of Novel CA within each of the 34 tissues expressing novel CAs.

**Figure 9 entropy-25-00319-f009:**
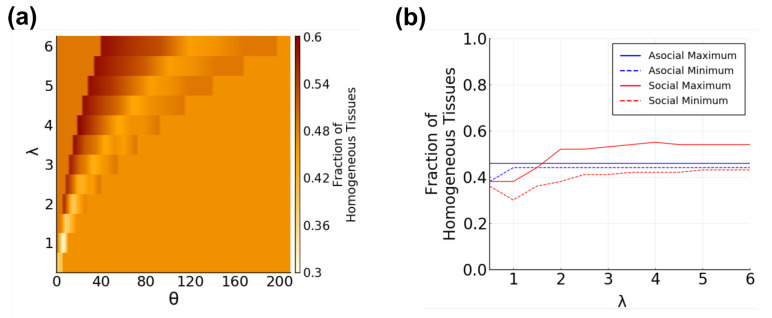
Fraction of tissue samples that are homogeneous as a function of λ and θ. (**a**) heatmap of fraction of homogeneous tissues as a function of λ and θ; (**b**) maximum (solid) and minimum (dashed) fraction of homogeneous tissues at each λ value, split by asocial (blue) and social (red) regions. Both (near) global max and global min occur in social regions at λ≤2.

## Data Availability

Not applicable.

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
