# Peer review of "Characterizing the Impact of Communication on Cellular and Collective Behavior Using a Three-Dimensional Multiscale Cellular Model"

_entropy, 2023, doi:10.3390/e25020319_

Round 1
Reviewer 1 Report
This is a superb and seminal paper on a central topic in biology. The authors study the effect of intercellular communication in a 3D model of a multicellular system. Each cell has a dynamically critical random boolean network of regulatory nodes, as well as receptor nodes and sensor nodes. The sensor nodes release diffusible molecules into the tissue. Above threshold concentration of these at a receptor activate the receptor.
The authors study the behaviors of the tissue in terms of two control parameters, lamba, the mean diffusion distance, and Omega, the threshold. They find 6 patterns, asocial and social. In the three social domains, they find the emergence of combination cell types, comprised of states on each of two different attractor cell types of the asocial networks. And they find entirely novel cell types. Obviously, the latter are candidates for the gradual evolution of new cell types in the history of life.
This paper is very well wrought and should surely be published. It is the first effort that I know of concerning three dimensional tissues. It will elicit and drive lots of further work, simulations then experiments. Please publish it as is
Reviewer 2 Report
This is a well-posed and clearly written theoretical treatment of the effect of cell communication on the development of a cluster of formal cells whose alternative states are determined by Boolean networks. It would be an important contribution to the theoretical and systems biology literatures and I support its publication. However, I have two distinct categories of criticism that I feel are essential to acknowledge and discuss before the paper is published.
The first point might stimulate additional simulations, but these would be optional, and the issue could be adequately handled by pointing to limitations of the analysis. The second point does not pertain to the simulations themselves, but to the applicability of the framework to developmental biology or other social behavior of cells. In my opinion, it needs to be discussed for the entire approach not to be misleading. The second point is therefore the more important one conceptually.
1. Signaling in multicellular clusters and aggregates may include lateral inhibition, whereby a cell signals ones adjacent to it to assume a different state from its own (this is typically mediated by Notch and one of its receptors in animals but can be by other gene products in other groups), and reaction-diffusion coupling, which can lead to Turing-type instabilities. The latter may employ juxtacrine lateral inhibition or other kins of negative feedback. It appears that the authors’ formalism would include such pattern-forming effects, but they are not manifested in the spatial arrangements produced by their simulations. These would typically include spatial periodicities of more than two peaks, such as those seen in hair follicles and limb digits. (It is possible that they are implicitly there, and I am missing them, but if so, their presence should be made more explicit by expression pattern diagrams rather than tables and graphs.) If the parameter choices used are too limited to show these important effects this should be indicated.
2. The use of the Boolean function paradigm for how gene regulatory networks determine differences between cell types via alternative dynamical attractors has been widely discussed and broadly accepted by theoretical biologists, but it conflicts with many experimental findings when used to characterize the full range of an organism’s cell types (see Cell differentiation: What have we learned in 50 years? - PubMed (nih.gov)). The main problem is that the closely coordinated cellular activities needed to build a coherently functional body or organ cannot simply be the mathematical consequences of a formal system. Further, developmental gene regulation by transcription factors (at least in animal systems, where enhancer-based expression hubs are utilized for this process) does not have the stable network topologies, stoichiometries and mass action required for global Boolean network determination of cell type. In a limited category of cases in animals, such as those described in A Computational Model of the Endothelial to Mesenchymal Transition - PubMed (nih.gov), Gene-free methodology for cell fate dynamics during development - PubMed (nih.gov) and Probabilistic boolean networks predict transcription factor targets to induce transdifferentiation - PubMed (nih.gov), where the transitions are among a few cell types adjacent to one another in a developmental lineage, and a common set of transcription factors are employed, the Boolean framework is relevant. There are similar cases in bacterial biofilms, where the constraint of functional coordination among the cell types is not present. This limitation of the Boolean paradigm must be discussed, in my view, for results of this study not to be misunderstood.
Reviewer 3 Report
This paper attempts to characterize the effects of cell-cell communication on the behavior of a tissue, including that of its component cells and that of the macroscopic tissue as a whole, using a simplified dynamic network model. While the goals of this paper are certainly important for the advancement of our understanding of multicellular systems, the framework that forms of the basis of all the analysis performed here may suffer from limitations that could affect the results. To that end, I would urge the authors to clarify those concerns and revisit the results and the surrounding discussion.
Major concerns:
- Defining the (a)social regimes by constraining the states of the neighboring cells to be the same is restrictive compared to allowing all possible combinations for all the cells. This is also likely the reason why almost 50% of the phase space is occupied by asocial regimes (fig 3a). If this restriction is relaxed, I suspect that the proportion of asocial regions may shrink by a factor proportional to the number of neighboring cells. To be precise, I think it would be equal to the proportion of the number of canalizing functions among the set of all possible Boolean functions of n+1 variables, where n is the number of neighboring cells. Therefore, I urge the authors to provide a justification for the choice of how the single-cell behavior is characterized in this paper, given the above context.
- How are A^T_ON,OFF,SELF, described in 3.2.1. computed? Is it done in the same way as fig 3a, except now with a GRN? Much of the analysis is based on comparing the Cas with these asocial CAs as the reference. Now, in the context of a tissue the CA of a cell is both a function of the state of R and the initial GRN states. Therefore, it's in principle possible that a cell enters the same CA as an asocial reference CA and yet the corresponding S-R logic does not follow asocial rules (A1, A2 or A3 in Fig.3b), by virtue of the history of the combined (S,R,GRN) states of the entire tissue. Therefore, I urge the authors to consider this possibility and either revisit the results or clearly state this as a limitation and discuss its potential implications.
- Since the analyses beginning 3.2.1 consider random regulatory networks, any observations must be attributable to a combination of communication parameters and the GRN characteristics, not communication alone. For example, with reference to the result mentioned at the top of p.9, could it be due to R not connecting to any of the regulatory nodes? In fact, the probability that R serves as an input among a set of 11 possible input nodes can be precisely calculated for k=3, so I wonder if that number could be 36%? Comment 3 follows a similar vein, except now the outcome may be due to a property of the initial state of the GRN. I would urge the authors to accommodate these points into how the results are interpreted.
- The usage of the terms "social cells" and "asocial cells" from section 3.2.1 onwards is confusing, as it could refer to either the (a)social cells within a tissue or an entire tissue (a collection of "cells"). The reference to the latter in actual usage within the text can only be inferred upon close reading. Likewise, the usage of the term "regions" could also be confusing, as it may refer to a region within a tissue or one in the phase space without further qualifications of the terms. In a similar vein, sometimes it's not clear what the authors specifically mean by "communication" -- is it the tissue connectivity, GRN connectivity, or the communication parameters lambda and theta, etc. I therefore urge the authors to clarify the language around these terms to avoid confusion.
- The paper may be greatly benefitted from more discussions of how to make sense of the results and what the implications are. Specifically, most of the results starting from fig.4 are cataloged but it's not discussed how to make sense of the same.
Other comments and questions:
- The result that signaling does not guarantee communication, while interesting, raises the question of why self-communication is not considered communication. Even otherwise, it may be argued that asocial behavior is actually a direct result of signaling with certain properties, and it's not to be construed as "no communication despite signaling".
- It may be helpful to the reader to understand fig.3a as a depiction of "steady state" behavior (though not devoid of concerns, as stated above), which is a standard dynamical systems analysis method.
- In fig.3a, how does one make sense of the observation that a large part of the space corresponding to large lambda values is asocial (A1) even with relatively large values of theta? Even though the math described in the SI is clear, it does not supply a clear intuition to understand this result.
- Likewise, how to make sense of the observation that the regimes could switch between social and asocial as one sweeps through the parameter space in certain directions? Moreover, this also suggests that the two categories are on par with each other in this sense, that is, one is not particularly more meaningful than the other.
- According to equation A15, there must exist a theta (fixed somewhere between theta2 and theta3) such that increasing lambda alone should lead to a switch from A2 to S2. But fig 3a suggests that such a switch is not possible without increasing both lambda and theta. How to reconcile this discrepancy in understanding? One possibility is that after theta2 and theta3 meet at the crossing point and switch their relative positions one of them switches directions -- but that would be a nonlinear behavior that the algebra of A15 doesn't warrant.
- Rather than the indicator function, wouldn't something like a Hamming distance between attractors be a more appropriate measure to assess the differences between attractors?
- How does the phase map change when regulatory nodes are included? Are there still the same 6 regimes? Are the social and asocial regions referred to in these discussions correspond to the original phase space or the new one where the regulatory networks are included?
- In fig.4a, The total number of tissues doesn't seem to add to 100; the number corresponding to b_lambda=0.0 is almost equal to 100. What do the blue steps represent? Also, please mark the social and asocial regions in these figures.
- The metrics corresponding to the rightmost two columns in fig.4a are defined much later. This could be confusing to some readers.
- With reference to the description at the bottom of p.11: aren't only three attractors possible for the asocial cells, namely those corresponding to ON, OFF and SELF?
- In fig.7, The horizontal axis M is not formally defined.
- In eqn,13, The term inside the indicator function should be CA_j = CA_i.
- Clarify how D_KL is computed when the cardinalities of P and Q could be different as it could be in this case, since the asocial reference always contained only three attractors? Also, how is Q computed here? Is it just a uniform distribution of 3 attractors?
- Some parts of the paper imply that the unit of lambda is the number of cells. Is this formally correct, as the formal definition of lambda doesn't seem to suggest it?
- Please make sure that the subscript notations in the math in the SI are correct.
Round 2
Reviewer 3 Report
I thank the authors for helping me better understand some of the technical details. While I'm satisfied with their responses, I would like to urge the authors to expand a little bit in the paper, if possible, in regard to the following. While it's now clear to me that the all-zeros and all-ones states of the neighboring cells define the points where the regime can switch, there's plenty of room for all sorts of behavior in the intermediate cases. The effect of this on the phase map is that it would add a gradation to it, making it look more like a heat map. Take region S1 for instance. For some of the parameter values, we can imagine truth tables (that incorporate all input combinations, not just the extreme cases) that would suggest behavior more akin to A2 rather than S1. In the ideal case, one would enumerate all input combinations (2^2046) and chart out all possible look-up tables and then compute exactly how representative the behavior S1 is for the entire region. That being not practical to pursue at this time, it may help to at least describe it as an aside. I feel that the reader would benefit from this deeper understanding of the actual phase space and be able to make better sense of all the results that follow in the paper.